# Utilizing Patient-Derived Epithelial Ovarian Cancer Tumor Organoids to Predict Carboplatin Resistance

**DOI:** 10.3390/biomedicines9081021

**Published:** 2021-08-16

**Authors:** Justin W. Gorski, Zhuwei Zhang, J. Robert McCorkle, Jodi M. DeJohn, Chi Wang, Rachel W. Miller, Holly H. Gallion, Charles S. Dietrich, Frederick R. Ueland, Jill M. Kolesar

**Affiliations:** 1Division of Gynecologic Oncology, Department of Obstetrics & Gynecology, University of Kentucky Chandler Medical Center, 800 Rose Street, Lexington, KY 40536-0263, USA; jwgo229@uky.edu (J.W.G.); raware00@uky.edu (R.W.M.); holly.gallion1@uky.edu (H.H.G.); charles.dietrich@uky.edu (C.S.D.); fuela0@uky.edu (F.R.U.); 2Department of Cancer Biostatistics, University of Kentucky Chandler Medical Center, 800 Rose Street, Lexington, KY 40536-0263, USA; zhuwei.zhang@uky.edu (Z.Z.); chi.wang@uky.edu (C.W.); 3Department of Pharmacy Practice & Science, University of Kentucky College of Pharmacy, 540 Healthy Kentucky Research Building, 760 Press Avenue, Lexington, KY 40539-0596, USA; rob.mccorkle@uky.edu; 4University of Kentucky College of Medicine, 800 Rose Street, MN150, Lexington, KY 40536-0298, USA; jde286@uky.edu

**Keywords:** ovarian cancer, tumor organoids, chemotherapy resistance, carboplatin, integrated genetic analysis

## Abstract

The development of patient-derived tumor organoids (TOs) from an epithelial ovarian cancer tumor obtained at the time of primary or interval debulking surgery has the potential to play an important role in precision medicine. Here, we utilized TOs to test front-line chemotherapy sensitivity and to investigate genomic drivers of carboplatin resistance. We developed six high-grade, serous epithelial ovarian cancer tumor organoid lines from tissue obtained during debulking surgery (two neoadjuvant-carboplatin-exposed and four chemo-naïve). Each organoid line was screened for sensitivity to carboplatin at four different doses (100, 10, 1, and 0.1 µM). Cell viability curves and resultant EC_50_ values were determined. One organoid line, UK1254, was predicted to be resistant to carboplatin based on its EC_50_ value (50.2 µM) being above clinically achievable Cmax. UK1254 had a significantly shorter PFS than the rest of the subjects (*p* = 0.0253) and was treated as a platinum-resistant recurrence. Subsequent gene expression analysis revealed extensively interconnected, differentially expressed pathways related to NF-kB, cellular differentiation (PRDM6 activation), and the linkage of B-cell receptor signaling to the PI3K–Akt signaling pathway (PI3KAP1 activation). This study demonstrates that patient-derived tumor organoids can be developed from patients at the time of primary or interval debulking surgery and may be used to predict clinical platinum sensitivity status or to investigate drivers of carboplatin resistance.

## 1. Introduction

Epithelial ovarian cancer (EOC) is the leading cause of death from gynecologic malignancy in the United States. In 2021, it is estimated that 21,410 women will be diagnosed with ovarian cancer and that it will be responsible for 13,770 deaths in the U.S. [1]. The high mortality rate is primarily due to the predominance of late-stage detection and the high rate of recurrence due to chemotherapy resistance.

The current standard of care treatment for advanced stage disease includes surgical debulking, with the goal of removing all macroscopic disease (R0 cytoreduction), in combination with platinum-based neoadjuvant or adjuvant chemotherapy [2]. Despite this aggressive front-line treatment, more than 80% of patients recur [3]. Historically, the platinum-free interval (PFI; time from the last dose of front-line adjuvant platinum-based chemotherapy to the detection of recurrence) has been used to classify patients into platinum-sensitive, -resistant, and -refractory groups [4]. These groups have considerably different clinical outcomes. “Platinum-sensitive” individuals (PFI ≥ 6 months) comprise approximately 75% of recurrences and have a median overall survival (OS) of 24–36 months. Conversely, 15% of patients are “platinum-resistant” (PFI < 6 months), with a median OS of only 9–12 months. Disease in “platinum-refractory” individuals progresses during treatment and makes up about 10% of recurrences. This group suffers the worst outcome, with a median OS of 3–5 months [5,6]. For clinicians, the choice of treatment in the platinum-resistant or -refractory setting is difficult. Response rates to non-platinum-based cytotoxic chemotherapy are similar and overall quite poor: topotecan (20%), gemcitabine (19%), liposomal doxorubicin (26%), oral etoposide (27%), docetaxel (22%), and weekly paclitaxel (21%) [2].

Despite the dramatic differences in outcomes between platinum-sensitive, -resistant, and -refractory groups, there is no validated method to predict clinical response to platinum-based chemotherapy, and all individuals receive the same up-front therapy. Chemosensitivity assays using patient-derived tumor cells have been increasingly explored to satisfy this unmet clinical need. In fact, some National Comprehensive Cancer Network (NCCN) centers employ the use of chemosensitivity assays to guide management in the face of recurrence when there are multiple equivalent chemotherapy options available [2]. Early investigations primarily using extreme drug resistance assays [7,8] or phenotypic drug response assays [9] were initially promising but have failed to produce sufficient evidence of efficacy to change the standard of care or warrant reimbursement [10]. However, patient-derived tumor organoid (TO) chemosensitivity assays have recently emerged as a more accurate model of in vivo tumor biology [11] and have shown promise to predict clinical chemotherapy response in vitro [12].

Here, we developed and validated six patient-derived epithelial ovarian cancer TO lines that were subsequently screened for sensitivity to front line standard of care chemotherapeutic agents. TO genetic sequencing was used to identify genomic determinants of carboplatin resistance. Our primary objective was to assess the ability of TOs to predict clinical outcomes to initial chemotherapy. Secondary objectives included the identification of an integrated genomic signature of platinum resistance in EOC.

## 2. Materials and Methods

### 2.1. Subjects

Women with suspected or histologically confirmed epithelial ovarian cancer with a plan to undergo cytoreductive surgery were eligible for study inclusion. All patients who were potentially eligible were approached for enrollment by trained clinical research staff during their pre-operative clinic visit, which occurred 1–4 weeks before the scheduled debulking surgery. Written informed consent was obtained from all subjects, and they were enrolled in the Total Cancer Care Protocol: A Lifetime Partnership with Patients Who Have or May be at Risk of Cancer (MCC 17-MTB-01, UK IRB #44224). Clinical outcome data were prospectively collected, deidentified, and correlated with TO chemosensitivity assay and genomic data by an honest broker. Disease assessments were performed per routine clinical practice by the treating provider to assess progression-free survival (PFS). Patient outcomes were followed until all patients demonstrated clinical evidence of recurrence or progression as defined by the RECIST version 1.1 criteria [13]. The study was conducted according to the guidelines of the Declaration of Helsinki, and it was approved by the Institutional Review Board of the University of Kentucky.

### 2.2. Tumor Organoid (TO) Development and Validation

Fresh ovarian tumor tissue was obtained from patients at the time of debulking surgery, dissociated into a single ovarian cancer cell suspension, and established in Matrigel^®^ Growth Factor Reduced Basement Membrane Matrix (Corning) in vitro using factor-defined media [14,15]. TOs were passaged at least two times to eliminate stromal cells by digesting the Matrigel^®^ matrix with trypsin-EDTA/TrypLE followed by gentle mechanical dissociation. Once ovarian cancer TOs were established, the TOs were fixed, and then representative sections were H&E stained and compared with a primary tumor by a board-certified pathologist.

### 2.3. Chemosensitivity Screens

Established TOs were enzymatically dissociated into single cells and plated in 384-well plates. The cells were cultured for 72 h prior to the administration of carboplatin at five different doses (0, 0.1, 1, 10, and 100 µM). After culturing for an additional 72 h, organoids were incubated with Hoechst nuclear counterstain and imaged on a spinning disc confocal high content imager. After imaging was completed, viability was measured by with an MTS assay (Promega). Raw data were generated in triplicate, and the average cell viability for each drug concentration was determined after normalizing values to untreated negative controls. Cell viability curves were generated, and EC_50_ values were determined. 

### 2.4. Sequencing Methods

The Tempus xT next generation targeted oncology sequencing assay was utilized to perform a gene mutation and expression analyses for all six generated TO cell lines. TO total nucleic acid was extracted and digested by proteinase K. RNA was purified from the total nucleic acid by DNase-I digestion. DNA and RNA sequencing was performed as previously described [16]. Briefly, 100 ng of DNA for each TO sample were mechanically sheared to an average size of 200 base pairs (bp) using a Covaris ultrasonicator. DNA libraries were prepared using the KAPA Hyper Prep Kit, hybridized to the xT probe set, and amplified with the KAPA HiFi HotStart ReadyMix. Next, 100 ng of RNA for each tumor sample were heat-fragmented in the presence of magnesium to an average size of 200 bp. Library preps were hybridized with the IDT xGEN Exome Research Panel, and target recovery was performed using streptavidin-coated beads, followed by amplification with the KAPA HiFi Library Amplification Kit. The amplified target-captured DNA tumor libraries were sequenced to an average unique on target depth of 500× on an Illumina HiSeq 4000. Samples were further assessed for uniformity, with each sample required to have 95% of all targeted bp sequenced to a minimum depth of 300× [17].

### 2.5. Gene Mutation and Gene Expression Bioinformatic Analysis

For somatic mutation analysis, an oncoplot was generated based on the maftools [18] package to visualize non-silent somatic mutations in DNA repair genes. For gene expression analysis, genes that were unexpressed or lowly expressed in all samples (no sample with counts per million mapped reads (CPM) > 1) were excluded from analysis. The differential expression analysis of the carboplatin-resistant versus carboplatin-sensitive groups was performed using the edgeR package [19]. Significantly differentially expressed genes were identified based on a threshold of false discovery rate (FDR) < 5% and annotated for gene ontology terms. A volcano plot was generated for results visualization. All these analyses were performed using R 4.0.3. The pathway enrichment and network analysis were performed using Qiagen’s Ingenuity Pathway Analysis (IPA) system for the core analysis of the RNA sequencing data and overlaid with the Global Molecular Network Overlay in the IPA knowledge base. 

### 2.6. Statistical Analysis

The classification of each TO cell line as carboplatin-sensitive or -resistant was based on the comparison of the carboplatin EC_50_ value to the clinically achievable plasma concentration of carboplatin. Resistant cell lines were defined as having a carboplatin EC_50_ above the plasma Cmax of carboplatin [20]. Sensitive TO cell lines had a carboplatin EC_50_ within achievable plasma concentrations. One sample *t*-test was used to compare carboplatin-resistant and pooled carboplatin-sensitive EC_50_ values using GraphPad Prism 8. The Kaplan–Meier method was used to estimate PFS curves for platinum-sensitive and -resistant patients. The PFS curves were compared via the log-rank test using R 4.0.3. A *p* < 0.05 was considered as statistically significant.

## 3. Results

### 3.1. Subject Demographic and Treatment Charactistics

The tumor samples used to generate the TO lines were derived from a relatively homogenous population. All subjects had histologically proven advanced-stage epithelial ovarian or fallopian tube cancer. Histologic subtype was exclusively high-grade serous. Primary disease sites were localized to the ovary (75%) and fallopian tubes (25%). One TO line (UK1393) was generated from a metastatic implant in the omentum, but all others were developed from the primary site of disease (Table 1). 

The treatment courses of the study subjects were also relatively homogenous. All participants were treated with a platinum and taxane doublet. One subject’s taxane therapy (UK1236) was switched from paclitaxel to abraxane due to allergic reaction. Most subjects were chemo-naïve (66.7%) at the time of debulking surgery. However, two subjects (33.3%) were exposed to three cycles of neoadjuvant chemotherapy before interval debulking surgery. Optimal cytoreduction was achieved in 50% of patients, and all other debulking surgeries achieved <0.5 cm of residual disease. Most patients did not receive maintenance therapy. However, one patient received olaparib, and another was enrolled in a Gynecologic Oncology Group (GOG) clinic trial studying the effects of the PARP inhibitor rucaparib and immunotherapy agent nivolumab [21]. It is uncertain if the patient received study drugs or placebo (Table 2). 

### 3.2. Tumor Organoid Validation

Histologic concordance between each ovarian cancer TO cell line and its respective primary ovarian cancer tumor sample was confirmed. After the establishment of the TO cell line, a sample of it was formalin-fixed and stained using hematoxylin and eosin (H&E) (Figure 1). Primary tumor samples for each established cell line were also formalin-fixed and H&E stained. The TO sample and respective tumor sample were compared. All TO lines were determined to be similar to their respective parental tumor samples after examination by a board-certified pathologist.

### 3.3. Chemosensitivity Screens

Cell viability curves and resultant EC_50_ values were determined for all generated TO lines (Table 3). 

The mean EC_50_ value for UK1254 exceeded achievable plasma carboplatin Cmax (50 µM) and was the highest of all TO line EC_50_ values. Conversely, all other TO lines were determined to be sensitive to carboplatin, with a significantly lower pooled cell viability EC_50_ mean value (*p* = 0.018). All carboplatin-sensitive TO cell lines demonstrated EC_50_ values within the range of achievable plasma concentrations (Figure 2).

### 3.4. Clinical Outcomes

The number of days from completion of adjuvant chemotherapy until recurrence or progression as demonstrated by RECIST criteria was used to determine each subject’s progression free survival (PFS) (Table 3). UK1254 had a significantly shorter PFS than the rest of the subjects with a *p* = 0.025 (Figure 3). Clinical outcomes directly correlate with TO cell viability chemosensitivity assay results.

### 3.5. Tumor Organoid Mutation Analysis

A limited mutation analysis was performed for all generated TO lines using the Tempus xT gene panel. Genes that were mutated in multiple cell lines and a selection of DNA repair genes were specifically interrogated to explore similarities in mutation profiles between TO cell lines for all enrolled subjects (Figure 4). As expected in high-grade serous ovarian cancer (HGSOC), the most commonly mutated gene was TP53 (4/6; 67%). The second most commonly altered genes were FANCC (2/6; 33%) which is a critical component of the Fanconi anemia core complex [22], and NOTCH2 (2/6; 33%), which is a key part of the Notch signaling pathway that controls the normal morphological development of multicellular organisms. Genomic mutations in the DNA repair genes were mostly relegated to intron alterations but also notably included a BRCA1 frame shift deletion mutation in UK2238, a BRIP1 missense mutation in UK1254, and an ATM missense mutation in UK1267. 

### 3.6. Tumor Organoid Gene Expression Analysis

We then performed a gene expression analysis using the RNA sequencing data in Qiagen’s IPA to evaluate differences in expression and pathways to better understand the mechanism of carboplatin resistance. All subjects had TO RNA sequencing data available. A total of 71 genes were significantly differentially expressed (FDR values < 0.05) between the carboplatin-resistant and carboplatin-sensitive TOs after appropriate thresholds were applied (Figure 5). Genes that met significance cutoff criteria are represented in blue. Genes eliminated after thresholds were applied are represented in red.

The top upregulated and downregulated differentially expressed genes comparing the carboplatin-resistant group to the carboplatin-sensitive group are displayed in Table 4. In the top ten upregulated genes, many are involved in transmembrane transport, cellular differentiation, and immune response modulation. In the top ten downregulated genes, many are involved in regulation of cellular growth, cellular stress response, and lipid metabolism. Notably, TMEM178B is not yet linked with an established biological pathway identifier and may represent a novel finding.

Next, an in depth network analysis used the RNA sequencing data to determine cell-specific pathways impacted by carboplatin resistance. Notably, leukocyte extravasation signaling, GP6 signaling, cardiac hypertrophy signaling, and PI3K signaling in B lymphocytes were predicted to be increasingly activated in the carboplatin-resistant TO, as demonstrated by pathways represented with shades of orange. Conversely, the CD40 signaling, HER-2 signaling, and MSP-RON signaling in macrophages were suspected to have decreased activation in the carboplatin-resistant phenotype and are represented in shades of blue (Figure 6A). Pathways that were differentially activated and downregulated were found to be extensively interconnected (Figure 6B).

To better assess the clinical applicability of the gene expression analysis, we converted the pathway analysis to a heatmap with analysis by disease and organ system (Figure 7A). The length of the box denotes the −log(*p*-value). The color of the boxes correlates with the z-score, with the intensity of blue representing *z* ≤ 0 and the intensity of orange representing *z* ≥ 0. Pathways related to organismal injury and abnormalities, cancer, gastrointestinal disease, and reproductive system disease predominated. This suggests that carboplatin resistance is partly mediated by the alteration of injury-associated biological mechanisms and well-established cancer-related pathways.

Finally, we performed network mapping using IPA with Global Network Overlay to explore the effect of carboplatin resistance on genes that were determined to be significantly altered between the carboplatin-resistant and carboplatin-sensitive groups. Upregulated expression is denoted in red, with the color intensity corresponding to increased significance. Conversely, downregulated expression is notated in green, with the color intensity again corresponding to increased significance. Network mapping results were filtered by statistically significant *p*-values with expression fold changes ≥ 0. We focused on the most significantly altered gene network (Figure 8A) and the second most significantly altered gene network (Figure 8B). Exploration of the most significantly altered network map (Figure 8A) revealed an interplay between various pathways all centered around NF-kB when the carboplatin-resistant TO was compared to the carboplatin-sensitive TOs. The second most significantly altered network (Figure 8B) demonstrates interplay between pathways involved in cellular differentiation (PRDM6 activation) and the linkage of B-cell receptor signaling to the PI3K–Akt signaling pathway (PI3KAP1 activation) [23]. 

## 4. Discussion

This prospective, observational, exploratory study demonstrates that TO development from chemo-naïve and neoadjuvant-chemotherapy-exposed epithelial ovarian cancer patients is both feasible and potentially predictive of clinical response to front line therapy. Commiserate with this study, other groups have successfully developed TOs from epithelial ovarian cancer patients and utilized TOs to screen for sensitivity to chemotherapeutic agents [24]. However, to our knowledge, this study is the first to report a prospective correlation of carboplatin chemo-sensitivity screening with PFS.

Our mutation analysis provides insight into the genetic underpinnings driving tumorigenesis in our population. As expected in high-grade serous ovarian cancer, alterations in the tumor-suppressor gene TP53 were the most common mutations. The second most commonly altered gene, FANCC, is a critical component of the Fanconi anemia core complex. A dysregulated Fanconi anemia pathway is frequently identified in epithelial ovarian cancer due to its extensive interconnection with DNA repair pathways [25]. Missense mutations in Notch2 occurred in two of the carboplatin-sensitive TO lines (UK1267 and UK1393). A wide range of cancer types have been found to overexpress Notch2 or to exhibit Notch2 gain-of-function mutations. Overactive Notch2 signaling has been linked to the dysregulation of certain miRNAs, tumor-associated stromal cell input, and the modulation of internal and external stimulation conditions in tumor cells that contribute to chemo- and radio-resistance [26]. If the Notch2 missense mutations identified in these ovarian cancer TO cell lines renders the Notch2 protein nonfunctional, then a dysregulated Notch2 signaling pathway may be partially responsible for the observed carboplatin sensitivity. An exploration of the mutation patterns of a selection of DNA repair genes reveals mostly intron mutations that likely do not impact function. However, notably, a BRIP1 missense mutation was identified in the resistant TO cell line UK1254, but the functional significance of this mutation is uncertain [27]. 

Though the gene panel mutation analysis provides some insight into the molecular drivers of tumor cell growth, it does not paint a complete picture of carboplatin resistance in UK1254. Our comparative gene expression analysis using TO RNA sequencing and IPA pathway analysis provides insight into the biological processes that are potentially driving chemotherapy resistance. An exploration of the most significantly altered network map (Figure 8A) revealed an interplay between various pathways all centered around NF-kB when the carboplatin-resistant TO was compared to the carboplatin-sensitive TOs. In addition to apoptosis threshold determination, the transcription factor NF-kB regulates multiple aspects of the innate and adaptive immune functions, and it serves as a pivotal mediator of inflammatory responses [28]. It has been well-established that various dysregulated signaling pathways can activate the NF-κB signaling pathway in ovarian cancer, which in turn promotes chemoresistance, cancer stem cell maintenance, metastasis, and immune evasion [29,30,31]. The second most significantly altered network (Figure 8B) demonstrates interplay between pathways involved in cellular differentiation (PRDM6 activation) and the linkage of B-cell receptor signaling to the PI3K–Akt signaling pathway (PI3KAP1 activation) [23]. These combined functions may be responsible for the observed clinical and in vitro carboplatin resistance of UK1254. The discovery of these cellular alterations provides novel insight into the mechanism of carboplatin resistance in UK1254 and may be able to be exploited with targeted therapy.

We found that in the top ten upregulated genes, many have been linked to platinum-based chemotherapy resistance (AQP1 [32,33] and RELN [34]), poor prognosis when exposed to platinum agents (LIPC [35] and FXYD2 [36]), or increased invasiveness (ADGRF2 [37]) when overexpressed. Notably, the upregulation of TMEM178B and ZNF723 has not been directly linked to carboplatin resistance, and understanding of their biological function in cancer remains limited. Interestingly, transmembrane protein 178B, the gene product of TMEM178B, has been identified as a novel downstream target of the nuclear factor kappa beta (NF-κB) ligand/phospholipase C gamma-2 signaling axis that modulates osteoclast activation [38]. NF-κB is a pleiotropic transcription factor key that determines the death threshold of cancer cells after exposure to platinum drugs and the inhibition of NF-κB sensitizes cells to the effects of platinum-based chemotherapy [30]. Thus, the overexpression of TMEM178B may produce a biological effect similar to the upregulation of NF-κB and warrants further investigation. Among the top ten downregulated genes, many have been linked to platinum-based chemotherapy resistance (MAPK1 [39,40], SLFN11 [41], and LYPD1 [42]), poor clinical outcome (ARNT2 [43]), or oncogenesis via the constitutive activation of wnt/*β* signaling (AXIN2 [44]) when under expressed. 

The main strength of our study was that we were able to successfully correlate TO chemosensitivity assay results with clinical PFS despite only including six subjects in the analysis and that we identified genomic predictors of response. These results contrast prior ovarian cancer tumor organoid publications that report in vitro sensitivity to antineoplastics but fail to include correlation with clinical outcome and genomic predictors of resistance. Genomic predictors of platinum resistance identified at the initial surgery have the potential to guide subsequent clinical management with the advantages of convenience and speed over organoid sensitivity testing [24,45,46,47]. The major limitation of our study was its small sample size. We intentionally only utilized advanced-stage, high-grade serous epithelial ovarian cancer specimens in an effort to create the most clinically and genetically homogenous sample possible. Though this strategy decreased the number of subjects eligible for inclusion in this study, it reinforces the clinical applicability of our results to this specific patient population with an unmet clinical need. In addition, focusing on this homogenous population limits the generalizability of our findings to other types of ovarian cancer. An additional limitation of this study was the lack of normal tissue organoid controls. At the time of debulking surgery, some subjects lacked normal human ovarian tissue due to complete destruction by malignancy. Thus, matched normal ovarian tissue was unavailable for culture.

The early stratification of patients into carboplatin-sensitive and -resistant cohorts, before clinical recurrence, may help delineate who should receive maintenance therapy with bevacizumab or a biosimilar. Furthermore, the combined use of TO chemosensitivity assay results and genomic markers of carboplatin resistance into a predictive scoring system of recurrence may provide a basis for additional cycles of cytotoxic chemotherapy beyond the traditional six. We envision that if the methodologies utilized here are applied to a larger cohort, we could develop a novel epithelial ovarian cancer predictive scoring system. The Oncotype DX test is a similar system that is currently the standard of care for adjuvant chemotherapy stratification in early stage, ER+, HER2/neu-negative breast cancer, and intermediate-risk prostate cancer. The development of an accurate scoring system that predicts an individual’s front line PFS has the potential to change the standard of care for high-grade serous ovarian cancer treatment and improve outcomes for thousands of patients every year. 

## 5. Conclusions

Tumor organoid (TO) development from chemo-naïve and neoadjuvant-chemotherapy-exposed epithelial ovarian cancer patients is both feasible and potentially predictive of clinical response to front line therapy. An integrated TO mutation and gene expression analysis can be utilized to investigate the molecular mechanisms of carboplatin resistance. The combination of these methods may provide the basis for development of a predictive recurrence scoring system that can be utilized to tailor maintenance and additional adjuvant therapy to individual patient needs.

## Figures and Tables

**Figure 1 biomedicines-09-01021-f001:**
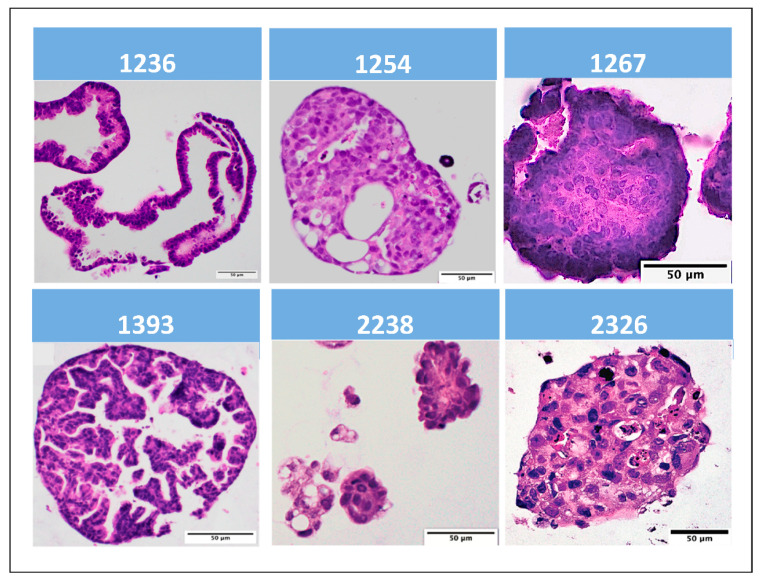
Hematoxylin and eosin (H&E) micrographs of established epithelial ovarian cancer tumor organoids.

**Figure 2 biomedicines-09-01021-f002:**
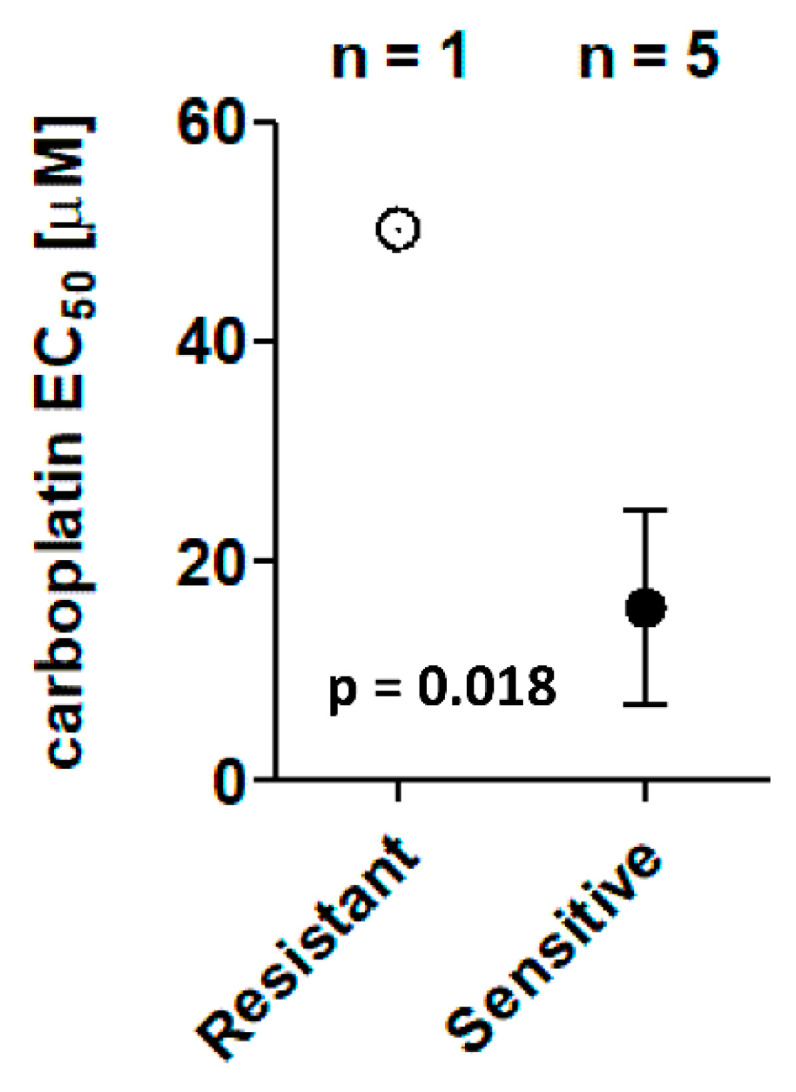
Scatter plot demonstrating cell viability EC_50_ mean ± SEM for the carboplatin-resistant (UK1254) and pooled carboplatin-sensitive TO lines when treated with carboplatin. One sample *t*-test was used to compare resistant and sensitive EC_50_ values (*p* = 0.018).

**Figure 3 biomedicines-09-01021-f003:**
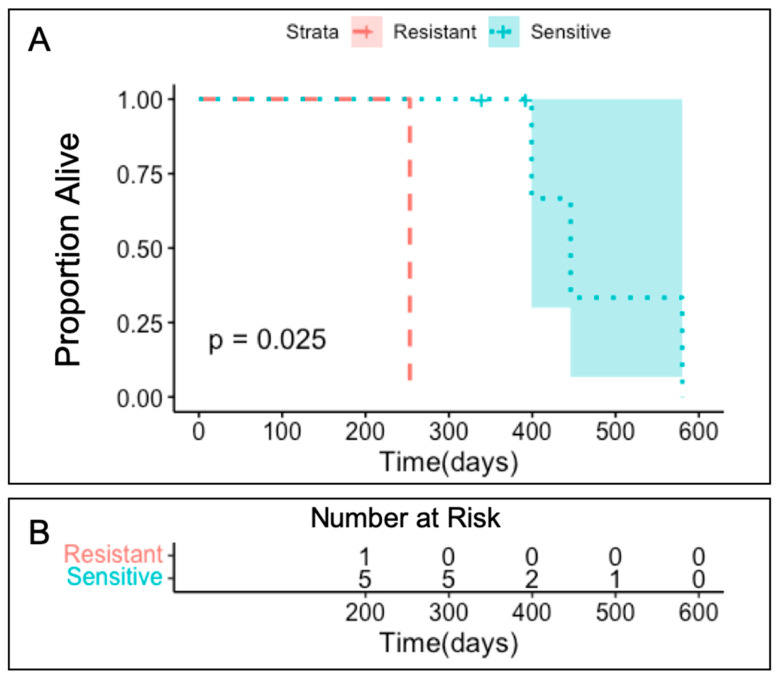
(**A**) Kaplan–Meier Survival plot demonstrating the PFS of enrolled subjects when stratified by carboplatin TO chemosensitivity assay results. Platinum-resistant (red): UK1254. Platinum-sensitive (blue): UK1236, UK1267, UK1393, UK2238, and UK2326. (**B**) Plot demonstrating number of subjects at risk for progression in platinum-resistant and -sensitive groups.

**Figure 4 biomedicines-09-01021-f004:**
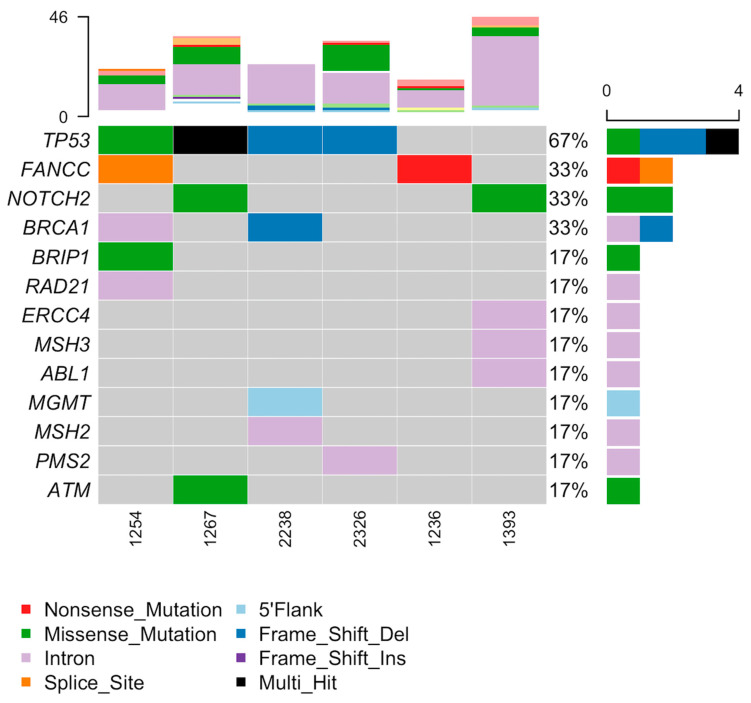
Oncoplot demonstrating the top mutated genes for all TO cell lines and select DNA repair genes. Carboplatin-resistant: UK1254. Carboplatin-sensitive: UK1236, UK1267, UK1393, UK2238, and UK2326.

**Figure 5 biomedicines-09-01021-f005:**
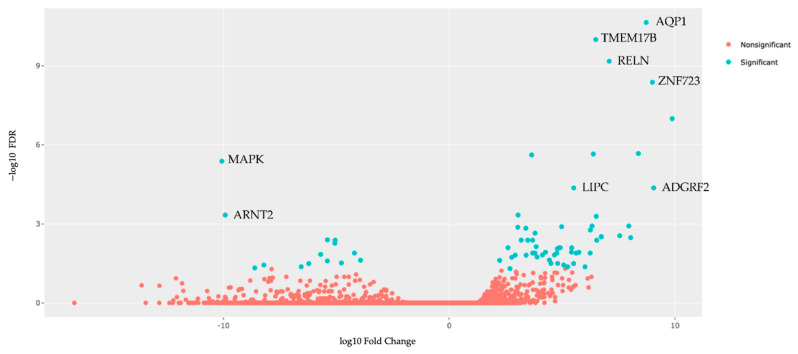
Volcano plot of the 71 differentially expressed genes identified when the carboplatin-resistant TO line (UK1254) is compared to the carboplatin-sensitive TO lines (UK1236, UK1267, UK1393, UK2238, and UK2326).

**Figure 6 biomedicines-09-01021-f006:**
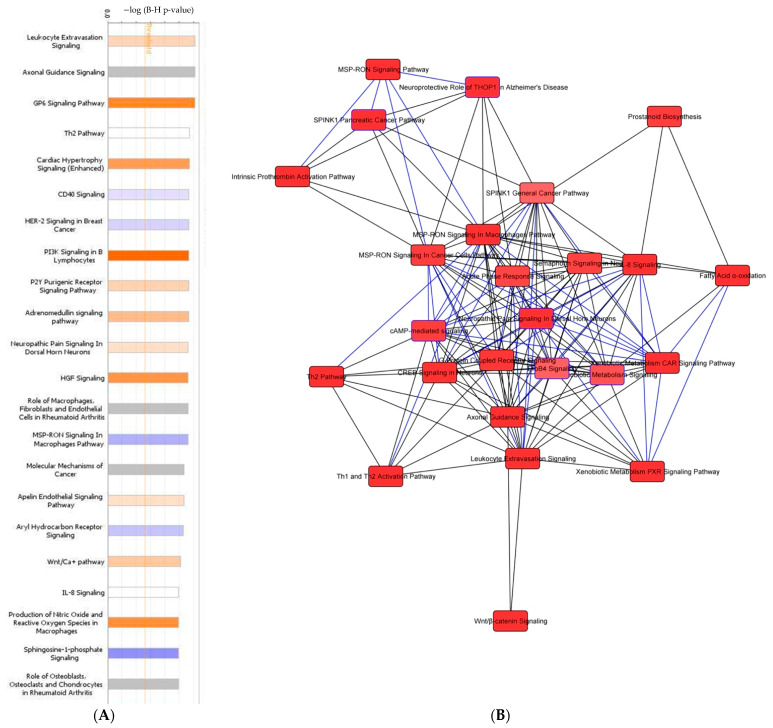
Differential expression analysis of carboplatin-resistant TO versus carboplatin-sensitive subjects. (**A**) Pathway analysis of genes differentially expressed between carboplatin-resistant and carboplatin-sensitive TOs. (**B**) Network analysis of the pathways differentially expressed between carboplatin-resistant and carboplatin-sensitive TOs.

**Figure 7 biomedicines-09-01021-f007:**
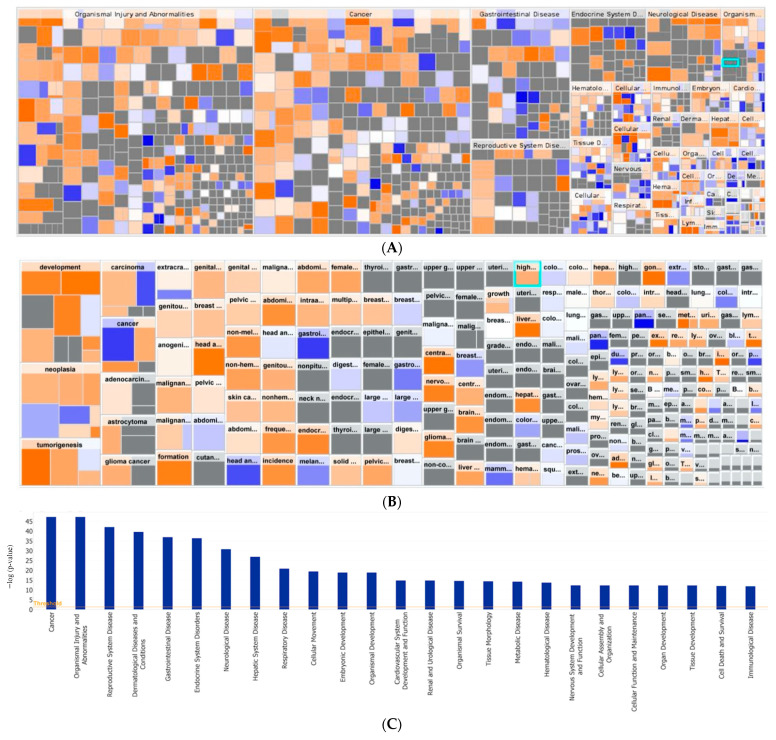
Gene network analysis between carboplatin-resistant and carboplatin-sensitive subjects. (**A**) Heatmap of the network analysis of genes differentially expressed between carboplatin-resistant and carboplatin-sensitive TOs by organ and disease system. The color and intensity of the boxes correlate with the z-score. Blue represents *z ≤* 0, and orange represents *z ≥* 0. (**B**) Heatmap of network analysis separated by cancer disease process. (**C**) Disease system pathways involved in carboplatin resistance are shown through network analysis of genes differentially expressed between carboplatin-resistant and carboplatin-sensitive TOs.

**Figure 8 biomedicines-09-01021-f008:**
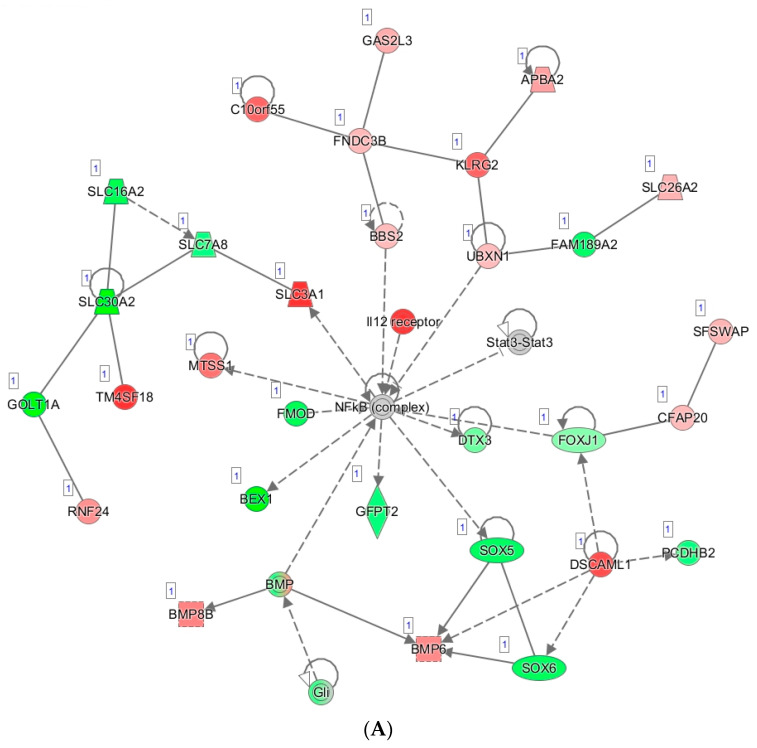
Network analysis of genes differentially expressed between carboplatin-resistant and carboplatin-sensitive TOs. Network mapping by Qiagen IPA with Global Network Overlay demonstrates the most significant gene network (**A**) and second most significant gene network (**B**).

**Table 1 biomedicines-09-01021-t001:** Demographic characteristics of included subjects.

ID	Age	TNM Stage	FIGO Stage	Primary Site	Histology	Grade
UK1236	48	ypT3cN0M1	IIIC	Ovary	Serous	3
UK1254	49	ypT3cNX	IIIC	Ovary	Serous	3
UK1267	55	T2bN0	IIB	Fallopian Tube	Serous	3
UK1393	46	T3cNX	IIIC	Ovary ^1^	Serous	3
UK2238	58	T3aN1b	IIIA	Fallopian Tube	Serous	3
UK2326	62	T3cNX	IIIC	Ovary	Serous	3

^1^ TO developed from metastatic omentum implant.

**Table 2 biomedicines-09-01021-t002:** Treatment courses of enrolled subjects.

ID	Residual Disease (cm)	Neoadjuvant	Adjuvant	Maintenance
UK1236	0	carboplatin and paclitaxel ^1^ × 1 cycle; carboplatin and abraxane × 2 cycles	carboplatin and abraxane × 3 cycles	none
UK1254	<0.5	carboplatin and paclitaxel × 3 cycles	carboplatin and paclitaxel × 3 cycles	GOG 3020: rucaparib v. placebo and nivolumab v. placebo
UK1267	0	None	carboplatin and paclitaxel × 6 cycles	none
UK1393	0	None	carboplatin and paclitaxel × 6 cycles; bevacizumab with cycles 2–6	none
UK2238	<0.5	None	carboplatin and paclitaxel × 6 cycles	olaparib
UK2326	<0.5	None	carboplatin and paclitaxel × 6 cycles	none

^1^ Carboplatin (AUC = 6) and paclitaxel (175 mg/m^2^) IV every 21 days was used as the standard dosing regimen.

**Table 3 biomedicines-09-01021-t003:** Cell viability EC_50_ values for each TO cell line when treated with carboplatin and subject progression-free survival (PFS).

ID	Carboplatin EC_50_ (µM)	PFS (Days)
UK2326	0.8	398
UK1267	1.1	338
UK2238	3.3	391
UK1236	28.5	579
UK1393	44.8	445
UK1254	50.2 ^1^	252

^1^ Above clinically achievable plasma Cmax.

**Table 4 biomedicines-09-01021-t004:** Top differentially expressed genes comparing the carboplatin-resistant group to the carboplatin-sensitive group. (**A**) Upregulated pathways in carboplatin-resistant TO compared to the carboplatin-sensitive TO group. (**B**) Downregulated pathways in carboplatin-resistant TO compared to the carboplatin-sensitive TO group.

**A. Upregulated**					
**Gene**	**LogFC**	***p* Value**	**QValue (FDR)**	**Pathway ID**	**Pathway Description**
1. AQP1	8.722968	1.46 × 10^−15^	2.26 × 10^−11^	GO:0022857	Transmembrane transport activity
2. TMEM178B	6.489275	1.30 × 10^−14^	1.01 × 10^−10^	---	---
3. RELN	7.083244	1.29 × 10^−13^	6.68 × 10^−10^	GO:0030154	Cell dedifferentiation
4. ZNF723	8.998623	1.08 × 10^−12^	4.20 × 10^−9^	GO:0003700	DNA Binding transcription factor activity
5. HAVCR1	9.870356	3.29 × 10^−11^	1.02 × 10^−7^	GO:00023676	Immune system process
6. FXYD2	8.374937	8.24 × 10^−10^	2.13 × 10^−6^	GO:0030234	Enzyme regulator activity
7. TGM3	6.3814	1.01 × 10^−9^	2.24 × 10^−6^	GO:0006464	Cellular protein modification process
8. OGFRL1	3.648762	1.25 × 10^−9^	2.41 × 10^−6^	GO:0007165	Signal transduction
9. LIPC	5.511889	2.91 × 10^−8^	4.31 × 10^−5^	GO:0006629	Lipid metabolic process
10. ADGRF2	9.051376	3.06 × 10^−8^	4.31 × 10^−5^	GO:0007165	Signal transduction
**B. Downregulated**					
**Gene**	**LogFC**	***p* Value**	**QValue (FDR)**	**Pathway ID**	**Pathway Description**
1. MAPK1	−10.0724	2.43 × 10^−9^	4.19 × 10^−6^	GO:0030154	Cell differentiation
2. ARNT2	−9.9226	3.89 × 10^−7^	0.000463	GO:0006950	Response to stress
3. STRA6	−5.39401	6.80 × 10^−6^	0.00405	GO:0006629	Lipid metabolic process
4. RBP1	−5.05496	8.05 × 10^−6^	0.004168	GO:0006629	Lipid metabolic process
5. ANTXR1	−5.06802	1.13 × 10^−5^	0.005453	GO:0007010	Cytoskeleton organization
6. LTBP1	−4.19728	3.59 × 10^−5^	0.01285	GO:0006464	Cellular protein modification process
7. AXIN2	−5.6952	4.47 × 10^−5^	0.01442	GO:0008283	Cell population proliferation
8. SLFN11	−3.93117	8.66 × 10^−5^	0.02396	GO:0006950	Cell response to stress
9. PHACTR1	−5.39943	9.58 × 10^−5^	0.025592	GO:0007010	Cytoskeleton organization
10. LYPD1	−4.77696	0.000116	0.030426	GO:0007267	Cell–cell signaling

## Data Availability

The data presented in this study are available on request from the corresponding author. The data are not publicly available due to concern for infringement on research subjects’ privacy.

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
