# Peer review of "Utilizing Patient-Derived Epithelial Ovarian Cancer Tumor Organoids to Predict Carboplatin Resistance"

_biomedicines, 2021, doi:10.3390/biomedicines9081021_

Round 1

Reviewer 1 Report

Patient-derived tumor organoids to evaluate the drug resistance is very common recently, and this submission hasn’t new insight or significance in this domain. Additionally, the quality of the data is poor and they can’t support the authors’ claim. Therefore, I personally suggest to decline this manuscript.

Author Response

Thank you for your comments. However, we respectfully disagree.  The major strengths of this manuscript are the selection of a homogenous population of patients, the comprehensive molecular & clinical annotation of enrolled subjects and the evaluation of organoids to predict clinical outcomes. Other recently published reports that utilize tumor organoids to investigate chemotherapy resistance in ovarian cancer do not include such a comprehensive translational investigation. For example:

  • Bi et al, tested 19 heterogenous ovarian organoids for sensitivity to common chemotherapy agents used for treating ovarian cancer, but present only a single case of clinical outcomes with no genomic profiling. PMID: 34200645.
  • Chen et al, report 6 ovarian organoids tested for sensitivity to investigational agents but provide no clinical correlations and report RNA Seq only. PMID: 32253045.
  • Nanki et al, report 7 ovarian organoids tested for sensitivity to common agents but no clinical outcomes and only somatic mutation profiling. PMID: 32724113.
  • Maenhoudt et al, reported on 3 organoids and sensitivity to common agents without genomic or clinical correlation. PMID: 32243841.

The discussion was revised as follows to address these key differences (additions are underlined).

“The main strength of our study is that we were able to successfully correlate TO chemosensitivity assay results with clinical progression free survival (PFS) despite only including six subjects in this analysis and that we have identified genomic predictors of response.  This is in contrast to prior ovarian organoids publications who report in vitro sensitivity to antineoplastics, but don’t include correlation with clinical outcome and genomic predictors of resistance [24, 43-45].  Genomic predictors of platinum resistance performed at initial surgery have the potential to guide subsequent clinical management with the advantages of convenience and speed over organoid sensitivity testing.”

In addition, the statement about being “the first to develop organoids from patients exposed to neoadjuvant chemotherapy” was removed, as Bi et al recently reported that they accomplished this feat.

Reviewer 2 Report

I would like to thank you very much for choosing me as a reviewer for a very interesting and innovative manuscript. I have a few minor reservations that may improve the manuscript.
- this sentence it must be change -
in primary surgery for ovarian cancer, the aim is to make the procedure as radical as possible in order to perform total debulking and not to perform cytoreduction with subsequent chemotherapy. As the current knowledge allows us to conclude that only primary surgery up to R0 is the best predictive factor for the patient

In addition, in the introduction, I would like to touch on the problem of how great a challenge to treatment are platinum-resistant patients. This is the group of patients most interested in these studies and the authors should highlight this problem In the discussion, I also miss a broader discussion (presenting data from several clinical trials) how difficult it is to choose chemotherapy for platinum-resistant patients Moreover, please add to the discussion the statistical data on the response to treatment in platinum-resistant patients in months

Author Response

I would like to thank you very much for choosing me as a reviewer for a very interesting and innovative manuscript. I have a few minor reservations that may improve the manuscript.

  • This sentence it must be change -in primary surgery for ovarian cancer, the aim is to make the procedure as radical as possible in order to perform total debulking and not to perform cytoreduction with subsequent chemotherapy. As the current knowledge allows us to conclude that only primary surgery up to R0 is the best predictive factor for the patient.

Author Reply: Thank you for your suggestion. We have changed the wording of this sentence (lines 43-45) to reflect your important insight. The sentence now reads:

“The current standard of care treatment for advanced stage disease includes surgical debulking with a goal of removing all macroscopic disease (R0 cytoreduction) in combination with platinum-based neoadjuvant or adjuvant chemotherapy [2].”

We have also changed the wording of the next sentence to clarify the methods of front line treatment so that the reader does not infer that the goal is to cytoreduce with chemotherapy. This sentence now reads:

“Despite this aggressive front-line treatment more than 80% of patients will recur [3].”

  • In addition, in the introduction, I would like to touch on the problem of how great a challenge to treatment are platinum-resistant patients. This is the group of patients most interested in these studies and the authors should highlight this problemIn the discussion. I also miss a broader discussion (presenting data from several clinical trials) how difficult it is to choose chemotherapy for platinum-resistant patients. Moreover, please add to the discussion the statistical data on the response to treatment in platinum-resistant patients in months.

    Author Reply: Thank you for your insight. We agree that this group of patients will be most interested in this study. We have added the following language to the discussion in the introduction section:

“For the clinician, the choice of treatment in the platinum resistant setting is difficult. Response rates to non-platinum based cytotoxic chemotherapy are similar and overall quite poor: topotecan (20%), gemcitabine (19%), liposomal doxorubicin (26%), oral etoposide (27%), docetaxel (22%), weekly paclitaxel (21%) [2].”

The reader is referenced back to the NCCN Ovarian Cancer guidelines which are complete with references to the clinical trials for each agent.

Additionally, the overall survival in months of the “platinum sensitive”, “platinum resistant” and “platinum refractory” groups are listed in lines 51-55. We feel that this paints the clinical picture for the reader.

Reviewer 3 Report

In the present study, the authors have utilized patient-derived epithelial ovarian cancer tumor organoids to predict carboplatin resistance. This is an interesting and informative translational study and covers very important aspects of oncology including therapy resistance and personalized medicine, however, there are a few issues that need to be addressed by the authors:

1-Although the authors have mentioned the reason for their small sample size, this is still problematic and underestimates one of the major clinical problems with EOC, which is tumor heterogeneity.

2-It could be interesting if the authors would have checked the sensitivity of the organoids to other chemotherapies and/or targeted therapies such as PARP inhibitors. How about a combination strategy in order to overcome carboplatin resistance in vitro? This could include inhibitors of PI3/AKT & NFKB signaling pathways and will confirm their gene expression analyses.

3-It has already been shown that NFKB has a key role in chemoresistance in epithelial ovarian cancer and its inhibition augments chemosensitivity. These research papers need to be mentioned and briefly discussed.

4-Fig. 7C, please increase the font size. The same with Fig. 6A

Author Response

In the present study, the authors have utilized patient-derived epithelial ovarian cancer tumor organoids to predict carboplatin resistance. This is an interesting and informative translational study and covers very important aspects of oncology including therapy resistance and personalized medicine, however, there are a few issues that need to be addressed by the authors:

1. Although the authors have mentioned the reason for their small sample size, this is still problematic and underestimates one of the major clinical problems with EOC, which is tumor heterogeneity.

Author Reply: We agree and have added the following to the limitations section (lines 390-392):

“In addition, focusing on this homogenous population limits generalizability of our findings to other types of ovarian cancer.” The issue of clonal selection and concern for lack of intratumoral heterogeneity in ovarian cancer tumor organoids has been previously addressed by Kopper et al (PMID: 31011202). This group found that “OC organoids recapitulate histological and genomic features of the pertinent lesion from which they were derived, illustrating intra- and interpatient heterogeneity”.

2. It could be interesting if the authors would have checked the sensitivity of the organoids to other chemotherapies and/or targeted therapies such as PARP inhibitors. How about a combination strategy in order to overcome carboplatin resistance in vitro? This could include inhibitors of PI3/AKT & NFKB signaling pathways and will confirm their gene expression analyses.

Author Reply: Thank you for this comment.  We agree this would be interesting but, this is somewhat outside the scope of this manuscript which is focused on identifying drivers of carboplatin resistance in order to predict clinical response in the front-line setting. The issue of overcoming carboplatin resistance via combination of cytotoxic and/or targeted chemotherapy is an important future direction and next step for this project.

3. It has already been shown that NFKB has a key role in chemo-resistance in epithelial ovarian cancer and its inhibition augments chemosensitivity. These research papers need to be mentioned and briefly discussed.

Author Reply:

Thank you. The following line has been added to the discussion section:

“It has been well established that various dysregulated signaling pathways can activate the NF-κB signaling pathway in ovarian cancer which in turn promotes chemoresistance, cancer stem cell maintenance, metastasis and immune evasion.” We have cited the following publications:

  • Harrington, B.S. and Annunziata, C.M. NF-κB Signaling in Ovarian Cancer. Cancers (Basel) 2019, 11(8):1182. doi: 10.3390/cancers11081182.
  • Lagunas, V.M. and Meléndez-Zajgla, J. Nuclear Factor-kappa B as a Resistance Factor to Platinum-Based Antineoplasic Drugs. Met Based Drugs 2008, 2008:576104. doi: 10.1155/2008/576104.
  • Kan, Y.; Liu, J. and Li, F. High Expression of Nuclear Transcription Factor-κB is Associated with Cisplatin Resistance and Prognosis for Ovarian Cancer. Cancer Manag Res 2020, 12:8241-8252. doi: 10.2147/CMAR.S265531

4. 7C, please increase the font size. The same with Fig. 6A

Author Reply: Done. The font size has been increased as much as possible but due to formatting constraints the size cannot be further increased. The zoom feature embedded in MDPI’s electronic journal format should allow readers to examine these figures in more detail if the manuscript is accepted for publication.

Round 2

Reviewer 1 Report

Thanks for the response from authors! However, I still do not think the results of this submission are significant. 

Author Response

Thank you for your comments. We appreciate your insight and believe that your comments have helped us improve the quality of this manuscript. However, we respectfully request the opportunity to publish our findings and allow the readers of Biomedicines the chance to assess the significance of the results.

Reviewer 3 Report

The authors have successfully addressed my previous comments/questions and I have no further comments to raise. 

Author Response

Thank you for your directed and helpful comments. We appreciate your insight and believe that your comments have dramatically helped us improve the quality of this manuscript.